# Dpp controls growth and patterning in Drosophila wing precursors through distinct modes of action

**Pablo Sanchez Bosch[1†], Ruta Ziukaite[2†], Cyrille Alexandre[2†], Konrad Basler[1], Jean-Paul Vincent[2*]**

[1]Institute of Molecular Life Sciences, University of Zurich, Zurich, Switzerland; [2]The Francis Crick Institute, London, United Kingdom

**Abstract** Dpp, a member of the BMP family, is a morphogen that specifies positional information in *Drosophila* wing precursors. In this tissue, Dpp expressed along the anterior-posterior boundary forms a concentration gradient that controls the expression domains of target genes, which in turn specify the position of wing veins. Dpp also promotes growth in this tissue. The relationship between the spatio-temporal profile of Dpp signalling and growth has been the subject of debate, which has intensified recently with the suggestion that the stripe of Dpp is dispensable for growth. With two independent conditional alleles of *dpp*, we find that the stripe of Dpp is essential for wing growth. We then show that this requirement, but not patterning, can be fulfilled by uniform, low level, Dpp expression. Thus, the stripe of Dpp ensures that signalling remains above a pro-growth threshold, while at the same time generating a gradient that patterns cell fates.

**\*For correspondence:** jp.vincent@
crick.ac.uk

[†]These authors contributed
equally to this work

**Competing interests:** The
authors declare that no
competing interests exist.

**Reviewing editor:** Utpal
Banerjee, University of California,
Los Angeles, United States

## Introduction

During development, tissue growth must be precisely coupled with patterning to ensure that the right number of cells can contribute to the various substructures within each organ (*Restrepo et al., 2014*) (*Baena-Lopez et al., 2012*; *Bryant and Gardiner, 2016*; *Hariharan, 2015*; *Irvine and Harvey, 2015*; *Johnston and Gallant, 2002*; *Wartlick et al., 2011a*). Not surprisingly, many signalling molecules that specify positional information also control growth (*Baena-Lopez et al., 2012*; *Restrepo et al., 2014*). This has been particularly well demonstrated in *Drosophila* wing imaginal discs, epithelial pockets that grow during larval stages and eventually give rise to the wing proper, the wing hinge and a part of the thorax called the notum (*Figure 1A*). Segregation of wing imaginal discs into the territories that give rise to these three structures is controlled by a series of signalling events involving EGFR, JAK/STAT, Notch, and Hedgehog signalling, culminating in sustained expression of Wingless and Dpp in orthogonal stripes until the end of the third instar (*Blackman et al., 1991*; *Neumann and Cohen, 1996*; *Zecca et al., 1995*). Both Wingless and Dpp are essential for growth (*Baena-Lopez et al., 2009*; *Burke and Basler, 1996*; *Restrepo et al., 2014*; *Spencer et al., 1982*; *Wartlick et al., 2011b*). Here, we focus on the role of Dpp, which is expressed along the anterior-posterior (A/P) compartment boundary in a pattern that cuts across the prospective notum, hinge and wing proper (*Figure 1A*). We look specifically at the prospective wing, which forms from a central region of the disc called the pouch. A wide range of evidence suggests that, in this region, Dpp acts as a morphogen. Graded distribution of the endogenous protein has not been directly visualized for lack of a suitable antibody against the mature secreted protein. However, the nested pattern of expression of target genes and the patterning activity of ectopic Dpp are strongly indicative of graded signalling activity (*Lecuit et al., 1996*; *Nellen et al., 1996*; *Schwank and Basler,*

**eLife digest** From the wings of a butterfly to the fingers of a human hand, living tissues often have complex and intricate patterns. Developmental biologists have long been fascinated by the signals – called morphogens – that guide how these kinds of pattern develop. Morphogens are substances that are produced by groups of cells and spread to the rest of the tissue to form a gradient. Depending on where they sit along this gradient, cells in the tissue activate different sets of genes, and the resulting pattern of gene activity ultimately defines the position of the different parts of the tissue.

Decades worth of studies into how limbs develop in animals from mice to fruit flies have revealed common principles of morphogen gradients that regulate the development of tissue patterns. Morphogens have been shown to help regulate the growth of tissues in a number of different animals as well. However, how the morphogens regulate tissue size and what role their gradients play in this process remain topics of intense debate in the field of developmental biology.

In the developing wing of a fruit fly, a morphogen called Dpp is expressed in a thin stripe located in the centre and spreads to the rest of the tissue to form a gradient. Bosch, Ziukaite, Alexandre et al. have now characterised where and when the Dpp morphogen must be produced to regulate both the final size of the fly's wing and the number of cells the wing eventually contains. The experiments involved preventing the production of Dpp in the developing wing in specific cells and at specific stages of development. This approach confirmed that Dpp must be produced in the central stripe for the wing to grow. Matsuda and Affolter and, independently, Barrio and Milán report the same findings in two related studies. Moreover, Bosch et al. and Barrio and Milán also conclude that the gradient of Dpp throughout the wing is not required for growth.

Further work will be needed to explain how the Dpp signal regulates the growth of the wing. The answer to this question will contribute to a better understanding of the role of morphogens in regulating the size of human organs and how a failure to do so might cause developmental disorders.

2010; *Zecca et al., 1995*) which is high around the A/P boundary, low further away, and undetectable at the lateral edges of the disc. High signalling activity, within and around the stripe of Dpp expression, is marked by immunoreactivity against phosphorylated Mad (P-Mad) and the expression of *spalt-major* (*salm*) while low signalling activity suffices to activate *optomotor blind* (*omb*) expression over a wider area of the prospective wing (*Burke and Basler, 1996*; *Lecuit et al., 1996*; *Nellen et al., 1996*; *Tanimoto et al., 2000*). In wing imaginal discs, Dpp signalling controls gene expression indirectly, through repression of a transcriptional repressor encoded by the *brinker* gene (*Martín et al., 2004*). Thus, the inverse gradient of Brinker expression provides yet another means of detecting Dpp signaling activity (*Schwank et al., 2008*).

As a morphogen, Dpp is a pattern organiser. For example, graded Dpp signalling determines the position of wing veins, particularly veins 2 and 5, through regulation of *salm* and *omb* (*Campbell and Tomlinson, 1999*; *Jaźwińska et al., 1999*; *Minami et al., 1999*). Dpp also clearly contributes to growth. Indeed, in the absence of Dpp signalling, wings (and other appendages) fail to grow (*Bangi and Wharton, 2006*; *Restrepo et al., 2014*; *Spencer et al., 1982*). The pro-growth role of Dpp is in part mediated through regulation of Myc (*Doumpas et al., 2013*), although a comprehensive understanding of growth regulation by Dpp signalling remains lacking. In wild-type imaginal discs, proliferation is approximately uniform while Dpp signalling is graded. Therefore, there is no apparent correlation between the level of Dpp signalling and the growth rate. How does a graded signal trigger a uniform response? Experiments involving the creation of abrupt differences in signalling suggested that local differences in Dpp signalling activity, that is, the spatial gradient of signalling, could be the trigger of growth (*Rogulja and Irvine, 2005*). This would provide an elegant mechanism for growth termination as the gradient would be expected to become shallower during growth (*Day and Lawrence, 2000*). However, there is no evidence that smooth differences in signalling activity associated with the endogenous gradient control growth. An alternative model is that the temporal gradient (the local relative increase in signalling activity) could be the trigger of

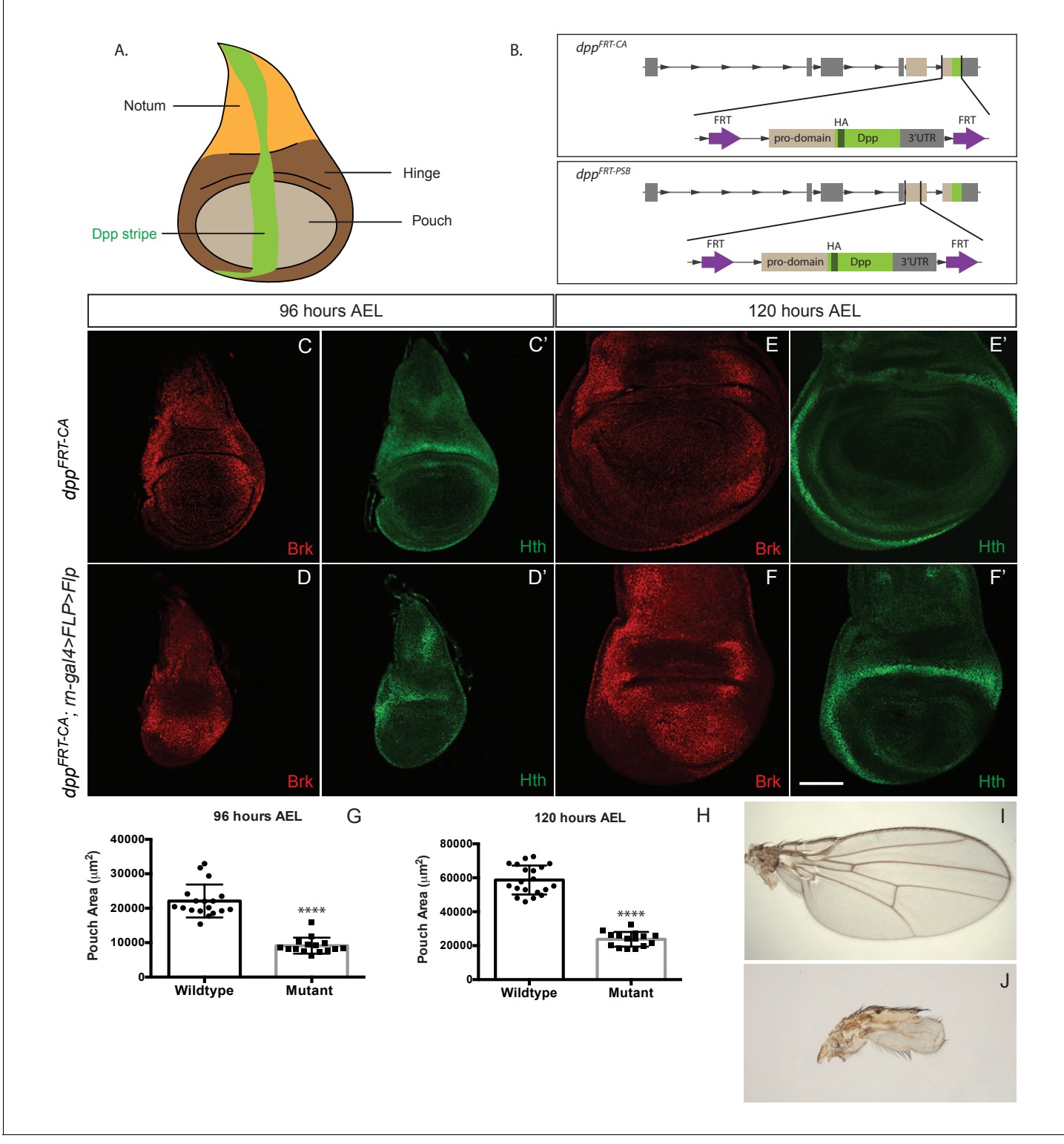

**Figure 1.** Growth of the prospective wing requires Dpp expression within the pouch. (**A**) Diagram highlighting the three domains of wing imaginal discs and the stripe of Dpp expression. (**B**) Diagram of the two conditional alleles we created, showing the region deleted from the genome and the inserted fragment. (**C–F'**). Inactivation of $dpp^{FRT-CA}$ in the pouch (with *rotund-gal4 UAS-Flp*) leads to derepression of *brinker* and reduced growth (shown here in discs fixed at 96 hr and 120 hr AEL). The edge of the pouch is marked by the weak inner ring of Hth expression. However, since the outer ring is more readily visible, this is the marker we used to measure pouch size (thus overestimating). (**G, H**) Quantification of the area enclosed by the Hth outer ring at the two stages (each dot/square represents one imaginal disc). (**I, J**) Wings from control (**I**) and experimental (**J**) adults. The scale bar, which

*Figure 1 continued on next page*

*Figure 1 continued*

represents 50 µm, applies to panels (**C-F′**). In panels (**G** and **H**) statistical significance of the difference between experimental and control samples was assessed with Student's t-test, assuming equal variance and a Gaussian distribution (p<0.0001).

The following source data and figure supplement are available for figure 1:

**Source data 1.** Pouch area.

**Figure supplement 1.** Inactivation of Dpp specifically in the pouch.

proliferation (*Wartlick et al., 2011b*), a model that has also been questioned (*Harmansa et al., 2015*; *Schwank et al., 2012*).

In agreement with the notion that Dpp controls growth through repression of *brinker*, imaginal discs lacking both Dpp and Brinker proliferate extensively (*Martín et al., 2004*; *Schwank et al., 2008*). Importantly, only the lateral region of the pouch (as well as the prospective hinge) overproliferates, while the medial area proliferate normally. Thus, depending on the distance from the stripe of Dpp, the cells of the pouch have a different propensity to proliferate. The main role of the Dpp/Brinker system would be to equalize this difference (*Schwank et al., 2008*). Thus, the inherent tendency of lateral cells to proliferate is slowed down by Brinker, while in medial cells Dpp emanating from its central stripe prevents Brinker-mediated suppression of growth.

Despite strong evidence in support of the above model, Akiyama and Gibson recently suggested that the central stripe of Dpp expression is dispensable for wing growth, and that the prospective pouch requires a source of Dpp in the anterior compartment to achieve growth (*Akiyama and Gibson, 2015*). To control Dpp activity, these authors created a conditional *dpp* allele (here referred to as $dpp^{FRT-TA}$) by deleting an essential exon and replacing it with a rescuing fragment flanked by Flp Recombination Targets (FRTs). They found that inactivation of this allele at the A/P compartmental boundary in the center of the medial region, had no adverse effect on growth. Inactivation was deemed effective within the pouch because no immunoreactivity against pro-Dpp was detectable there. This led the authors to conclude that the central stripe of Dpp, from where the Dpp gradient originates, is not required for growth. To account for the continued growth observed in the absence of the Dpp stripe, they suggest that perhaps low level Dpp originating from the anterior compartment could suffice to promote growth in the pouch. Here we show, with two new validated conditional alleles, that deletion of the central stripe of Dpp is deleterious to growth. We then investigate and compare the requirements of Dpp within the pouch for growth versus patterning.

## Results

### Wing growth requires Dpp expression in the prospective wing

To generate means of reliably controlling Dpp activity, we devised two conditional *dpp* alleles, $dpp^{FRT-CA}$ and $dpp^{FRT-PSB}$, that can be inactivated by Flp (*Figure 1B*). In both cases, hemaglutinin (HA) tags were included to enable detection of endogenously produced mature Dpp. Flp was then expressed in various patterns to trigger excision of the essential exon. First, Dpp production was inactivated throughout the prospective wing either with *rotund-gal4* and *UAS-Flp* in homozygous $dpp^{FRT-CA}$ or with *nubbin*-Gal4 and *UAS-Flp* in homozygous $dpp^{FRT-PSB}$. No HA immunoreactivity (HA-Dpp) could be detected in the pouch from 96 hr after egg laying (AEL) onward (*Figure 1—figure supplement 1*), indicating efficient gene inactivation. HA (i.e. Dpp) was still detectable in the prospective hinge and notum, as expected since Gal4 activity was mostly confined to the pouch. Immunostaining with anti-Brinker showed that *brinker* expression was derepressed throughout the pouch (*Figure 1C–F* and *Figure 1—figure supplement 1*), confirming that Dpp signalling was eliminated there. Note that the down-regulation of Brinker around residual Dpp expression in the hinge did not extend into the pouch (arrowhead in *Figure 1—figure supplement 1D*), suggesting that Dpp produced in the hinge has little effect on gene expression in the pouch. In both experiments, growth was markedly impaired, an effect that was quantified for $dpp^{FRT-CA}$ by marking the edge of the pouch with anti-Homothorax (anti-Hth) (*Azpiazu and Morata, 2000*; *Casares and Mann, 2000*)

and measuring the enclosed area at 96 and 120 hr AEL (*Figure 1G–H*). The pouch of experimental discs (*dpp^FRT-CA*; *rotund-Gal4, UAS-Flp*) was significantly smaller than that of their wild-type siblings at equivalent stages. It was, however, not completely eradicated, perhaps because of delayed *dpp* inactivation or residual BMP signalling by *glass bottom boat* (*gbb*) (*Ray and Wharton, 2001*). Since the *dpp^FRT-CA*; *rotund-Gal4, UAS-Flp* genotype is viable, the growth deficiency was also readily apparent in the adults that emerged (*Figure 1I,J*). These results confirm that production of mature Dpp within the pouch is required for this tissue to grow and that Dpp originating from outside the pouch does not compensate.

## Temporal requirement of dpp for wing growth

To assess whether Dpp is continuously required for wing growth, we first inactivated *dpp^FRT-PSB* at different times by Flp expressed from a *hsp70-Flp* transgene. Larvae were heat shocked at 48, 72 and 96 hr AEL and wing imaginal discs were fixed at 120 hr AEL. Staining with anti-HA confirmed the efficiency of gene inactivation although occasional spots of residual HA-Dpp expressing cells could be detected (*Figure 2*). Inactivation of *dpp* at 48 and 72 hr AEL resulted in widespread derepression of *brinker*, confirming the impairment in Dpp signalling. Heat shocking at 48 and 72 hr AEL resulted in markedly reduced growth, while later excision (96 hr AEL) had a milder effect. The

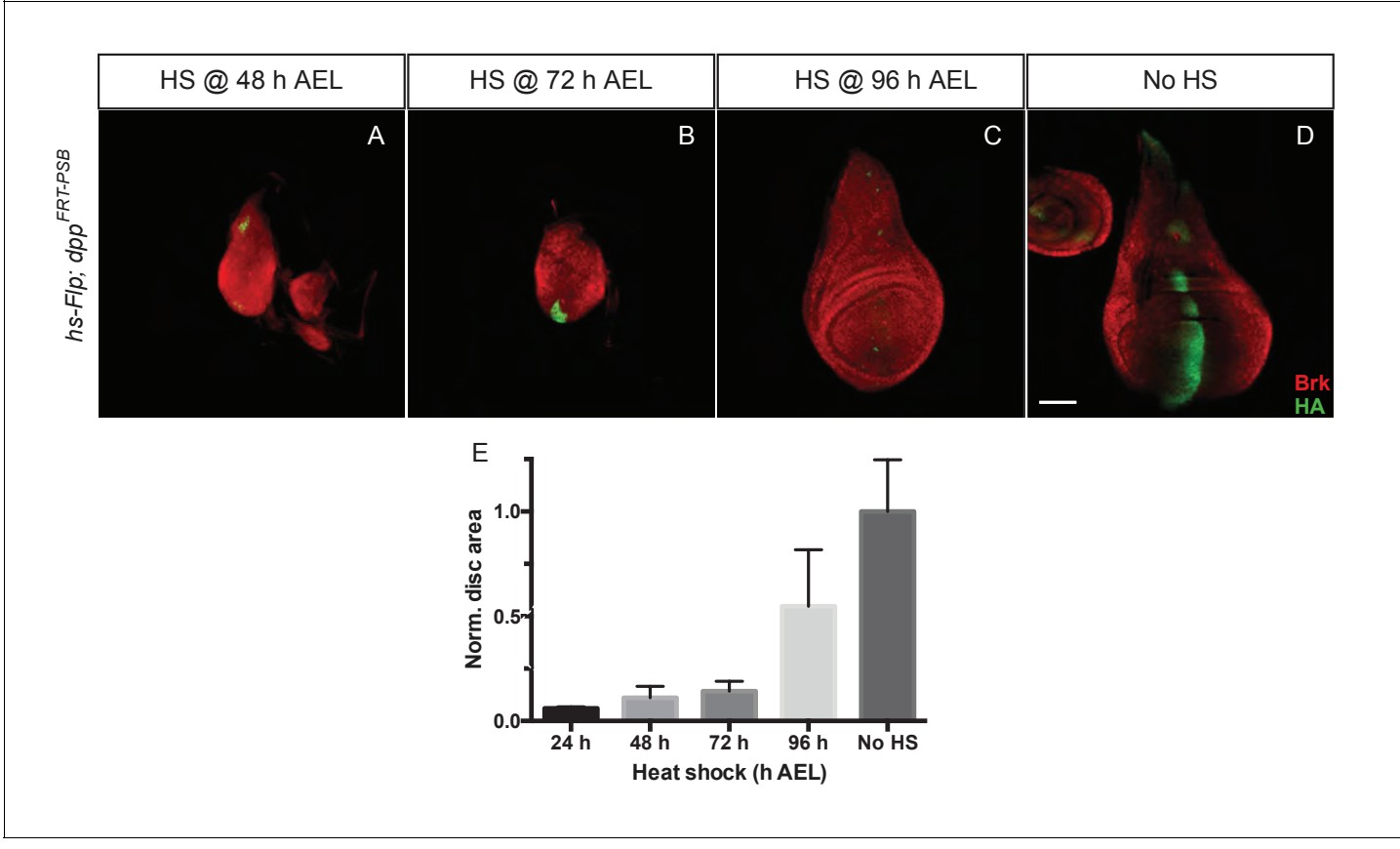

**Figure 2.** Temporal requirement of Dpp for growth. (**A–D**) Imaginal discs at 120 hr AEL following inactivation of *dpp^FRT-PSB* by induction of *hsp70-Flp* at the indicated times. Inactivation of Dpp leads to ubiquitous derepression of *brinker*, with the exception of residual HA-Dpp expressing clones (Representative examples are shown). (**E**) The total surface area of discs heat shocked at 24, 48, 72, and 96 hr AEL was measured and normalised to the average surface area of control discs (n = 4 for 24 hr AEL and n = 20 for the other time points). Area measurement for each time point was compared to the control area (no heat shock) with a one-way ANOVA. The p value was highly significant (<0.0001) for every side by side comparison except for 96 hr AEL vs 120 hr AEL. Scale bar = 50 μm.

The following source data is available for figure 2:

**Source data 1.** Total disc area.

relatively weak impact of heat shocks at 96 hr could be due to perdurance of Dpp or downstream events. Alternatively, any effect on growth might be hard to detect beyond this time because the growth rate of imaginal discs decreases with age (*Johnston and Sanders, 2003*). We conclude that the results of timed inactivation experiments show that Dpp must be continuously produced at least up to 96 hr, perhaps beyond, for the prospective wing to grow.

## Wing growth requires the endogenous stripe of dpp expression

Our findings so far indicate that Dpp must be produced in the pouch and during the 48–96 hr AEL period in order for the wing to grow. In this region, the major expression domain of Dpp is in a stripe along the A/P boundary (*Masucci et al., 1990*). It is therefore expected that, as shown in *Figure 3*, inactivation of Dpp specifically in this stripe would eradicate Dpp expression in the pouch and lead to growth impairment. Surprisingly, inactivation of $dpp^{FRT-TA}$ with Flp expressed under the control of dpp-Gal4 ($dpp^{FRT-TA}$ $dpp^{BLK}$-Gal4 UAS-Flp) was reported to have no adverse effect on growth (*Akiyama and Gibson, 2015*). In this genetic background, expression of *salm* and *omb* was disrupted, indicating that Dpp production was indeed impaired. It was therefore suggested that the stripe of Dpp expression may not be needed for growth because of the existence of another source of Dpp outside the stripe (*Akiyama and Gibson, 2015*). Indeed, long-term lineage tracing by G-TRACE suggests that progenitors of cells located anterior to the stripe could express Dpp (*Evans et al., 2009*), at least at some point during development. To gain further information on the pattern of *dpp* expression in the wing pouch, we created a reporter line ($dpp^{FRT-REP}$) expressing the readily detectable marker CD8-GFP from the endogenous *dpp* locus. An excisable cassette expressing Dpp was included upstream of the CD8-GFP coding sequences (*Figure 3—figure supplement 1A*) to allow expression of functional Dpp during embryogenesis, which requires two functional alleles. Thus, during embryogenesis, CD8-GFP is not expressed and the two alleles produce wild-type Dpp. Only after expression of Flp does this allele act as a reporter, in the domain of Flp expression. Cassette excision was induced after embryogenesis with *rotund-Gal4* and *UAS*-Flp, making CD8-GFP a reporter of *dpp* transcription in the pouch. At 72, 96 and 120 hr AEL, GFP was only detectable along the A/P boundary (*Figure 3—figure supplement 1B–D*). Thus, anterior to the stripe, the activity of the *dpp* promoter must either be very low or take place before 72 hr AEL. Therefore, it is unlikely to promote growth, at least after this time period. This conclusion spurred us to re-assess the role of the Dpp stripe in growth.

We tested the role of the endogenous stripe of Dpp in wing growth by inactivating our conditional alleles with *UAS-Flp* and *dpp-Gal4*. To enable comparison with the results of Akiyama and Gibson (*Akiyama and Gibson, 2015*), we chose the same $dpp^{BLK}$-Gal4 transgene (*Staehling-Hampton et al., 1995*). This strain was generated many years ago and kept separately in our respective laboratories. We therefore characterised the different $dpp^{BLK}$-Gal4 lines by splinkerette PCR (*Potter and Luo, 2010*). Although the three stocks displayed sequence polymorphisms, they all carried the $dpp^{BLK}$-Gal4 transgene at the same location, confirming that they all originated from the same initial stock and could be used interchangeably (*Figure 3—figure supplement 2*). The $dpp^{BLK}$-Gal4 UAS-Flp combination was introduced in $dpp^{FRT-CA}$ and $dpp^{FRT-PSB}$ homozygotes to inactivate *dpp* within the stripe. In both cases, efficiency of excision was assessed by staining imaginal discs with anti-HA, which marks functional, mature Dpp in the unexcised alleles. At 96 hr AEL, HA immunoreactivity was eliminated from the whole disc, except in a previously characterised zone located outside of the pouch, in the posterior prospective hinge (*Foronda et al., 2009*) (arrowhead in *Figure 3B,D* and *Figure 3—figure supplement 3C,D*). Such residual expression is reproducible and likely represents an area where $dpp^{BLK}$-Gal4 does not recapitulate the endogenous Dpp expression domain, as noted previously (*Akiyama and Gibson, 2015*). However, in the rest of the disc, including the whole pouch, the $dpp^{BLK}$-Gal4 UAS-Flp combination appeared to trigger efficient recombination and hence inactivation of *dpp*. Importantly, this was associated with derepression of *brinker* (*Figure 3B,D*) and a marked reduction (84%) of pouch size at the end of the growth period (*Figure 3G* and *Figure 3—figure supplement 3E*).

The lack of growth noted above is in contrast with the report that $dpp^{FRT-TA}$ $dpp^{BLK}$-Gal4 UAS-Flp imaginal discs attain a normal size and express Brinker throughout the pouch at 120 hr AEL (*Akiyama and Gibson, 2015*). This is in stark contradiction with the model that Dpp stimulates growth through repression of Brinker and that Brinker expression in the pouch is incompatible with growth (*Schwank et al., 2008*). To investigate this apparent inconsistency, we re-examined $dpp^{FRT-}$

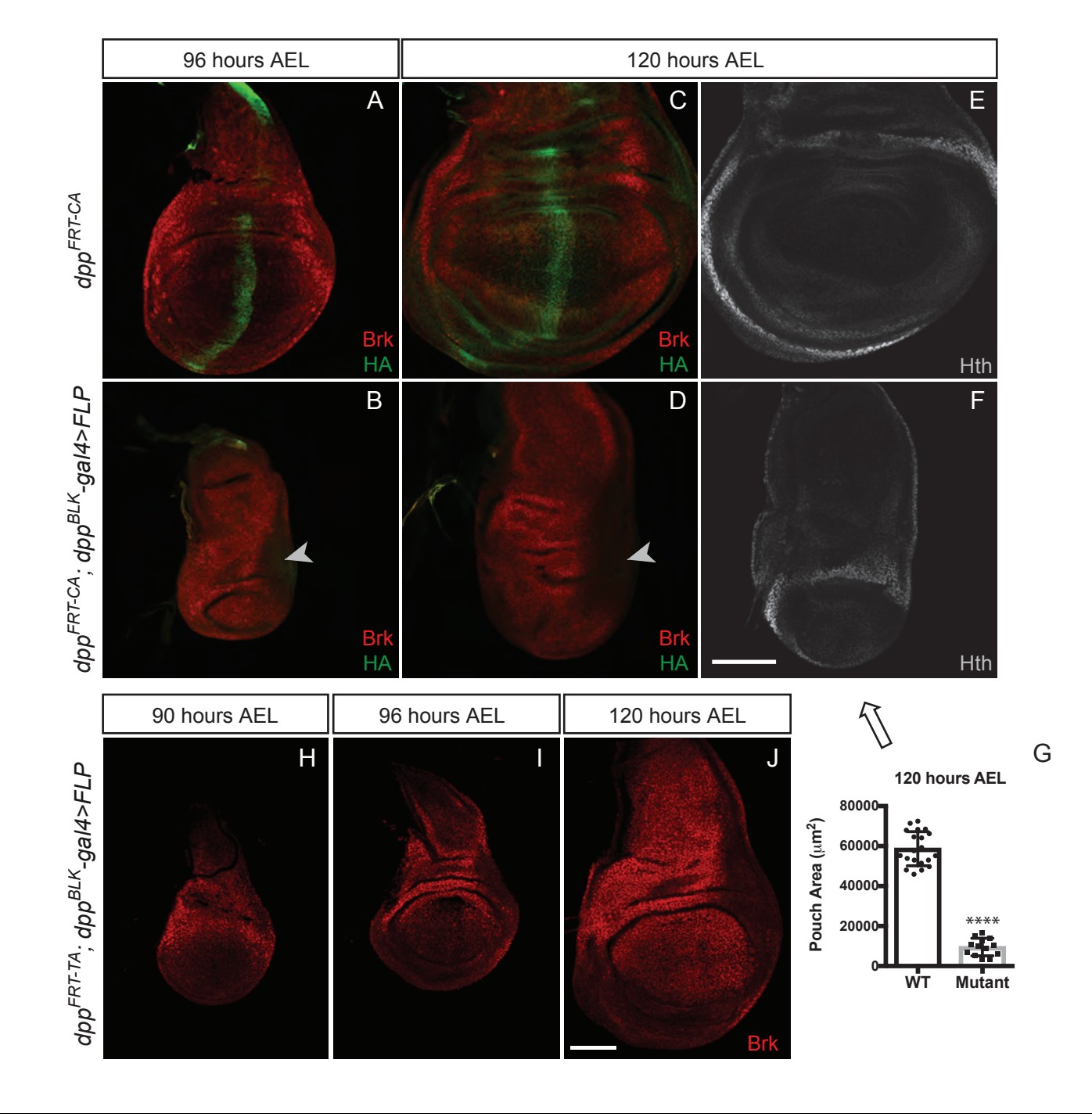

**Figure 3.** Growth of the prospective wing requires the endogenous stripe of Dpp expression. (A–F) Inactivation of $dpp^{FRT-CA}$ in the normal domain of Dpp expression (with $dpp^{BLK}$-Gal4 UAS-Flp) leads to depression of *brinker* and reduced growth (shown here in discs fixed at 96 and 120 hr AEL). A zone of *brinker* repression can be seen in the prospective hinge around weak residual Dpp expression (arrowhead in **B**, **D**). (**G**) Quantification of the pouch area (area enclosed by the outer ring of Hth) in control and experimental discs (each dot/square represents a disc). Asterisks in panels G denote the statistical significance of the difference between experimental and control samples, using Student's t-test, assuming equal variance and a Gaussian distribution. (**H–J**) Inactivation of $dpp^{FRT-TA}$ in the normal domain of Dpp expression (with $dpp^{BLK}$-Gal4 UAS-Flp) only leads to Brinker derepression after growth has taken place. At earlier stages (90 and 96 hr AEL), Brinker is repressed, indicating residual Dpp signaling activity. Scale bar = 50 μm.

The following source data and figure supplements are available for figure 3:

*Figure 3 continued on next page*

*Figure 3 continued*

**Source data 1.** Pouch area.
**Figure supplement 1.** A reporter inserted at the locus shows that *dpp* expression is confined to the stripe along the A/P boundary.
**Figure supplement 2.** Comparison of various *dpp-Gal4* strains.
**Figure supplement 3.** Inactivation of *dpp*$^{FRT-PSB}$ in the domain of *dpp* expression abolishes growth.

$^{TA}$ *dpp*$^{BLK}$*-Gal4 UAS-Flp* imaginal discs, not only at 120 hr AEL but also at earlier stages. We confirmed that the discs attain a normal size and express Brinker at 120 hr AEL (*Figure 3J*). However, at 90 and 96 hr AEL, during the growth phase, Brinker was repressed within the pouch (*Figure 3H,I*), a clear indication that Dpp signalling is still active at these stages. We suggest that, in this genotype, Dpp signalling is eradicated but only after most growth has taken place. These results suggest that the TA allele may not be as readily inactivated by *dpp*$^{BLK}$*-Gal4 UAS-Flp* as the PSB and CA alleles.

The efficacy of gene inactivation was assessed for all three alleles by expressing Flp from a *hs-Flp* transgene under identical heat-shock conditions and measuring *brinker* expression by qRT-PCR. The results show that *brinker* expression was derepressed in all cases but less so with *dpp*$^{FRT-TA}$ than with *dpp*$^{FRT-PSB}$ and *dpp*$^{FRT-CA}$ (*Figure 4A*). These results indicate that *dpp*$^{FRT-TA}$ is less readily excised than the other two alleles. Allele 'excisability' was also assessed functionally by measuring imaginal disc size following heat-shock-induced expression of Flp at different times (*Figure 4B–K*). Growth was impaired in a more pronounced manner with *dpp*$^{FRT-PSB}$ and *dpp*$^{FRT-CA}$ than with *dpp*$^{FRT-TA}$, especially with a heat shock at 72 hr AEL, a time when inactivation of Dpp signalling has a strong effect on growth (see quantification in *Figure 4K*). Therefore, molecular and functional assays suggest that the *dpp*$^{FRT-TA}$ allele may not be as readily inactivated as our alleles, perhaps because of differences of sequence context around the FRT sites. We note that one of the FRTs of *dpp*$^{FRT-TA}$ is flanked by a LoxP site, which could conceivably impair recombination. In any case, our results show that precluding striped expression of Dpp along the A/P boundary does interfere with wing growth.

## Uniform Dpp expression suffices for growth but not patterning

Our results so far show that Dpp expression from the endogenous stripe is required for the growth of wing precursors. They do not address, however, whether a spatial or temporal gradient is necessary. To investigate this question, we took advantage of our conditional alleles to eliminate endogenous *dpp* expression while at the same time inducing uniform constant expression from a transgene. The *rotund-Gal4* and *UAS-Flp* combination was used to simultaneously excise the FRT cassettes of *dpp*$^{FRT-CA}$ and *Tubα1-FRT-f*$^+$*-FRT-dpp*, a transgene previously shown to trigger intermediate signalling activity, sufficient to activate *omb* but not *salm* expression (*Zecca et al., 1995*). As expected, in the resulting 'rescued' discs, Omb was expressed uniformly, although at a reduced level and Brinker was repressed. (*Figure 5A–D*). However, pMad immunoreactivity was at the low level normally seen in the lateral region (*Figure 5E,F*), suggesting that the level of signalling achieved by *Tubα1-dpp* is similar to that present far from the normal stripe of Dpp. About half the discs of this genotype reached an approximately normal size at the end of the third instar while the other half overgrew slightly (as is the case for the disc shown in *Figure 5B*). Sustained growth was confirmed by assessing proliferation rates with anti-pH3 staining of discs dissected from late larvae crawling in the food. As shown in *Figure 5I–L*, 'rescued' and wild-type discs proliferated at approximately the same rate while discs lacking *dpp* proliferated at a lower rate in the pouch area. This result suggests that uniform and constant Dpp signalling is sufficient to promote growth in the pouch. It also suggests that the level of signalling needed to promote growth is much lower than that needed to produce peak p-Mad immunoreactivity.

Since veins form at stereotypical positions in Drosophila wings, they provide a convenient marker of patterning. The five longitudinal veins are distinctly specified by various signalling pathways (reviewed in [*Blair, 2007*]). Most relevant for this paper, the positioning of veins 2 and 5 is dependent on Dpp signalling. Prospective veins can be recognised in late imaginal discs as zones of *DSRF* (*Drosophila* serum response factor) repression (*Montagne et al., 1996*; *Nussbaumer et al., 2000*).

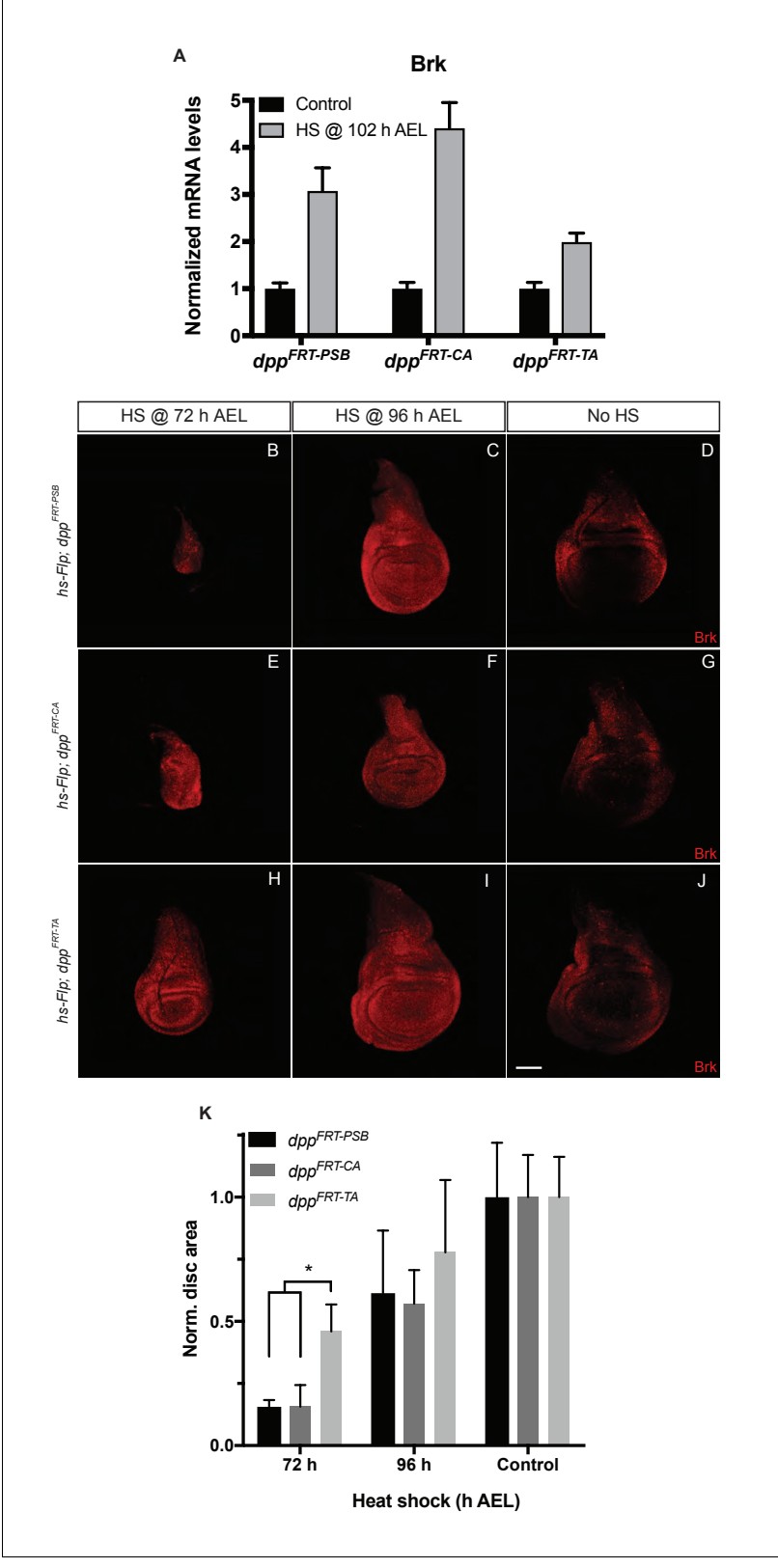

**Figure 4.** Inactivation efficiency for three conditional alleles of *dpp*. (**A**) Efficiency of inactivation for *dpp*<sup>FRT-PSB</sup>, *dpp*<sup>FRT-CA</sup> and *dpp*<sup>FRT-TA</sup> by Flp expressed from *hsp70-Flp* induced at 102 hr AEL. Level of *brinker* mRNA, normalized to that in non-heat-shocked controls, was assessed by qRT-PCR at 120 hr AEL. Each bar shows average mRNA level +/- SEM. A two-way ANOVA test showed statistically different *brinker* expression between *dpp*<sup>FRT-PSB</sup> and *dpp*<sup>FRT-TA</sup> (p=0.0041) as well as between *dpp*<sup>FRT-CA</sup> and *dpp*<sup>FRT-TA</sup> (p<0.0001). (**B–J**) Imaginal discs of the same genotypes were fixed and stained

*Figure 4 continued on next page*

*Figure 4 continued*

with anti-Brinker at 120 hr AEL, following a heat shock at 72 or 96 hr AEL or in the absence of heat shock (control). As can be seen, the 72 hr heat shock did not impair growth as much in *dpp^FRT-TA* as it did in *dpp^FRT-PSB* and *dpp^FRT-CA*. (K) Quantification of disc surface area (normalized to average surface area of control discs) at 120 hr AEL for the nine conditions shown in panels (B–J). Each bar represents data for 10 discs. Asterisk denotes statistical significance, as assessed by a two-way ANOVA test (p=0.029). Scale bar = 50 μm.

The following source data is available for figure 4:

**Source data 1.** Primers for qPCR.
**Source data 2.** Normalised Brk mRNA levels.
**Source data 3.** Total disc area.

Staining with anti-DSRF showed that the prospective vein pattern was markedly disrupted in 'rescued' discs (*Figure 5G,H*), with only two zones of repressed *DSRF* remaining, one around the D/V boundary, where vein 1 normally forms under the control of Wingless (*Couso et al., 1994*; *Rulifson and Blair, 1995*), and one around pro-veins 3 and 4, which are specified by Hedgehog in the wild type (*Blair, 2007*). The areas of DSRF repression corresponding to veins 2 and 5 were conspicuously missing. Because some of the 'rescued' larvae survived to adulthood, we were able to further assess, in adult wings, the extent of growth and patterning that uniform Dpp promotes. A majority of these wings appeared to be made entirely of crumpled vein material (*Figure 5O*), which made it difficult to assess size. This phenotype can be explained by the vein-specifying role of Dpp in pupal wings (*Sotillos and de Celis, 2006*). Nevertheless, a minority of 'rescued wings' were remarkably well formed (*Figure 5N*), perhaps because they experienced lower Dpp signalling at the pupal stage, below the threshold for vein specification. In these wings, vein patterning was disrupted, but reproducibly so, with a broad swath of vein tissue forming near the A/P boundary. Crucially, these wings reached a remarkably large size (compare *Figure 5M and N*). This result suggests that uniform, low level Dpp signalling promotes near-normal growth although this is not adequate for patterning.

## Discussion

Dpp behaves as a classic morphogen in wing imaginal discs of *Drosophila*. It is produced from a stripe of cells along the A/P boundary and spreads from there to activate the nested expression of target genes, which in turn position longitudinal veins. In addition to providing patterning information in the prospective wing, Dpp also promotes growth via repression of *brinker*. How graded Dpp signalling leads to homogenous proliferation has been the subject of discussion but until recently, there has been general agreement that the stripe of Dpp is required for growth. This basic tenet was recently challenged with a conditional *dpp* allele that can be inactivated in time and space by Flp (here referred to as *dpp^FRT-TA*). Inactivation in the normal domain of Dpp expression, with Flp driven by a disc-specific *dpp* regulatory element, was reported to have minimal impact on growth (*Akiyama and Gibson, 2015*). The authors suggested that Dpp expressed from a source in the anterior half of the pouch could suffice to sustain growth. Consistent with this suggestion, inactivation of *dpp* throughout the pouch with *nubbin-Gal4 UAS-flp* led to strong growth reduction (*Akiyama and Gibson, 2015*), an observation that we confirmed with our conditional alleles (*dpp^FRT-CA* and *dpp^FRT-PSB*) and two pouch-specific sources of Flp. However, inactivation of our alleles with *dpp^BLK-Gal4 UAS-Flp* (the same source of Flp used by *Akiyama and Gibson, 2015*) led to a severe impairment in growth (*Figure 3* and *Figure 3—figure supplement 3*), in contrast to the finding with *dpp^FRT-TA*. Our analysis of *brinker* expression during the growth period in the various mutant backgrounds allows us to reconcile the apparent discrepancy between our data and those of *Akiyama and Gibson (2015)*. We suggest that our alleles (*dpp^FRT-CA* and *dpp^FRT-PSB*) are more readily inactivated than the one generated by *Akiyama and Gibson (2015)* (*dpp^FRT-TA*). Thus, in the *dpp^FRT-TA*; *dpp-Gal4 UAS-Flp* genotype, cells expressing Dpp within the stripe would linger long enough to provide sufficient signalling activity for *brinker* repression (*Figure 3H,I*) and hence growth. As time goes on,

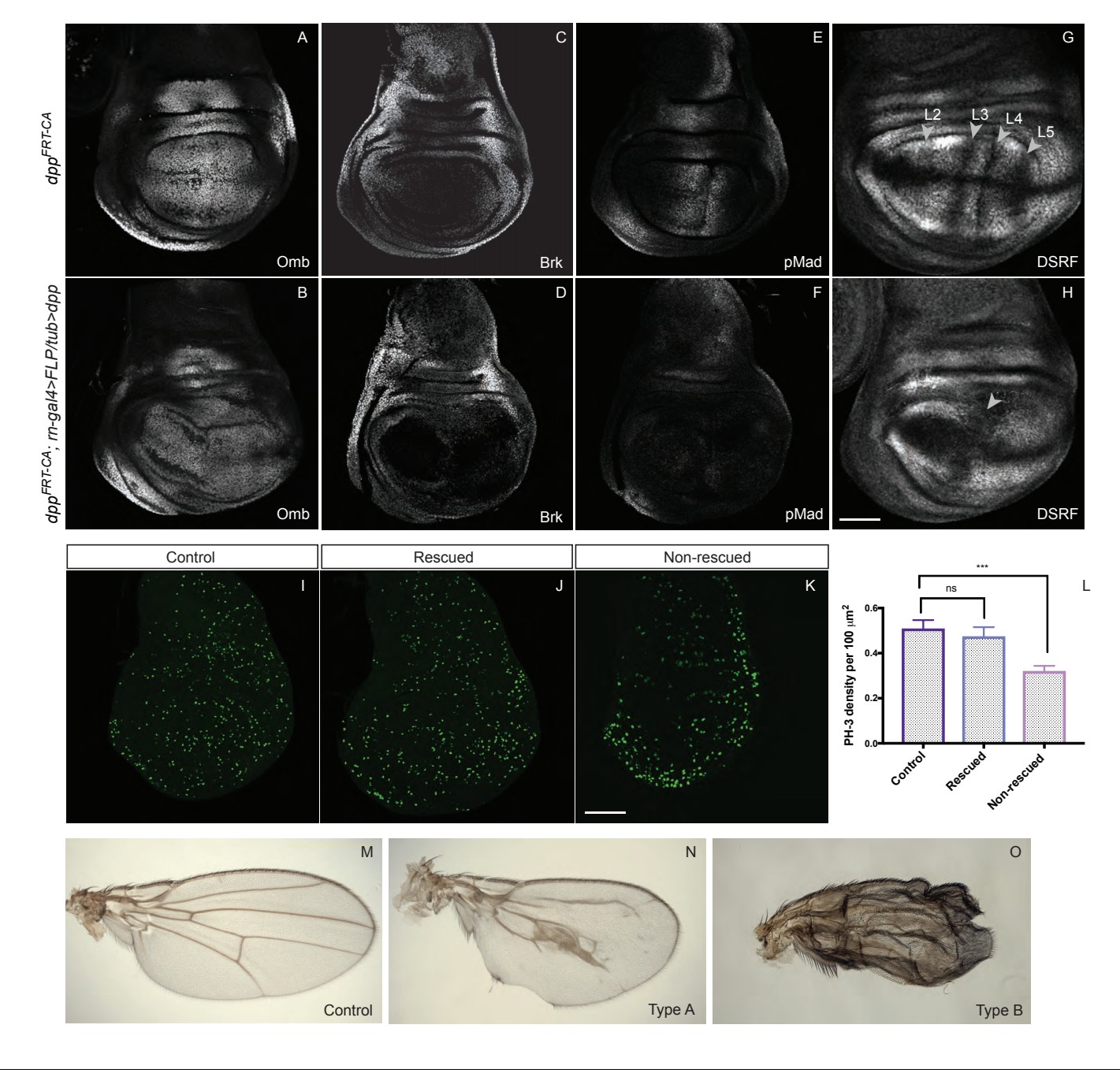

**Figure 5.** Low level uniform Dpp expression suffices for growth but not patterning. (**A–H**) Comparison of wild-type discs (**A, C, E, G**) to discs lacking endogenous Dpp in the pouch and expressing weak uniform Dpp instead (**B, D, F, H**). Uniform Dpp allows discs to reach a relatively normal size, although with a variably deformed shape (representative examples are shown). Omb is expressed in experimental discs, an indication of active Dpp signaling, but at a relatively lower level than in control discs (samples shown in A and B were stained and imaged under identical conditions). Note also the repression of Brinker and the loss of pMad expression in experimental discs. In contrast to their relatively normal size, experimental discs show abnormal vein patterning, with only two vein territories recognizable instead of the normal five (marked by the absence of DSRF immunoreactivity) (**G, H**). (**I–K**) pH3 immunoreactivity shows that, in control and rescued discs, proliferation is sustained seemingly normally (**I, J**) while proliferation in the pouch of non-rescued discs is depressed (**K**) Quantification show in L is based on 14 rescued discs, 9 controls and 11 unrescued discs. Statistical significance was assessed with a Student's t-test, assuming equal variance and a Gaussian distribution. Mitotic density (pH3 spots/area) was determined for each individual disc using a code written in Fiji (see *Figure 5—source data 1*). (**M–O**) Wings from the above genotypes. A majority of examined experimental wings (15/20) had excess vein tissue (**O**) while the remainder (5/20) had one central vein around the position of the A/P boundary and another (not visible) along the margin (**I**). Each micrograph is representative of 7–10 discs. Scale bar = 50 μm.

*Figure 5 continued on next page*

*Figure 5 continued*

The following source data is available for figure 5:

**Source data 1.** PH3 density.

these lingering cells would progressively undergo excision so that at the end of third instar, no signalling would remain, explaining the widespread derepression of *brinker* seen at the late 120 hr AEL stage (*Akiyama and Gibson, 2015*). Since, with our conditional allele, inactivation of Dpp in the endogenous stripe leads to growth impairment, we conclude that, during normal development, this source of Dpp is needed for growth, although as discussed below, this can be overcome with low-level exogenously expressed Dpp.

How does the Dpp gradient emanating from the Dpp stripe promote growth? Our finding that uniformly expressed Dpp is sufficient for growth suggests that a spatial gradient of signalling is not required. Moreover, the *tubulin* promoter, which was used to drive uniform expression, is expected to be constant over time. Therefore, our result could be taken as evidence against the model that growth depends on continuously rising signalling activity (*Wartlick et al., 2011b*), although it could be argued that even under a condition of uniform expression, signalling could rise if Dpp became more stable over time. Nevertheless, we prefer the simple model whereby, in the prospective wing, Dpp signalling over a threshold would be permissive for growth. The level of this threshold is still to be precisely measured. In the experiment illustrated in *Figure 5*, growth rescue by uniform Dpp in the pouch correlates with repression of *brinker*, consistent with the growth equalization model (*Schwank et al., 2008*). Although Akiyama and Gibson showed that *dpp*$^{FRT-TA}$ *dpp*$^{BLK}$-*Gal4 UAS-Flp* discs express *brinker* uniformly at 120 hr AEL (*Akiyama and Gibson, 2015*), as we have shown (*Figure 3H,I*), *brinker* only becomes derepressed in this genotype after growth has occurred. The observations that Dpp expression from the *Tubα1-dpp* transgene (*Figure 4*) or residual Dpp from a few cells within the stripe (as we propose is occurring in the *dpp*$^{FRT-TA}$ *dpp*$^{BLK}$-*Gal4 UAS-Flp* background), stimulate growth suggest that relatively low level signalling suffices for growth throughout the pouch (i.e. the prospective wing). As we have shown, this level of signalling is below that needed to produce substantial pMad immunoreactivity but higher than that needed to repress *brinker*. Better tools to tune the level of Dpp signalling will be needed to assess the relationship between signalling activity and growth at all stages.

Our results have significantly clarified the spatial requirement of Dpp. As we have shown, Dpp must originate from the pouch for this tissue to grow: in several experimental conditions (*Figure 3B, D*, *Figure 1—figure supplement 1C–F*, *Figure 3—figure supplement 3C–D*), Dpp produced outside the pouch could not overcome the absence of Dpp within the pouch. We cannot discriminate at this point whether the boundary between these tissues acts as a barrier to the spread of Dpp or whether these sources of Dpp are too weak to have an impact in the pouch. In any case, these observations confirm our assertion that growth is normally sustained by Dpp produced at the A/P boundary. Dpp signalling above a relatively low threshold is permissive for growth within the pouch throughout wing development. For this activity, the signalling gradient is irrelevant. By contrast, the signalling gradient is essential for patterning as it specifies the domains of *salm* and *omb* expression and thus the positions of veins. Thus, the dual role of Dpp in growth and patterning requires that it is expressed in a stripe. Late inactivation of Dpp impairs patterning, suggesting that the gradient information could be read at the end of the growth period. It remains to be determined how the two processes - growth and patterning - are coordinated to ensure the reproducible formation of the adult wing.

## Materials and methods

### Drosophila strains

Two conditional *dpp* alleles, illustrated in *Figure 1B*, were created for this study. In one allele, *dpp*$^{FRT-CA}$, the exon encoding mature Dpp was deleted and replaced with the same sequence flanked by FRT and modified so that it would encode two HA tags downstream of the three furin

cleavage sites. For the other allele, $dpp^{FRT-PSB}$, a portion of the first coding exon including the signal sequence was replaced by a FRT-flanked fragment encoding full-length HA-tagged Dpp (3xHA tag). See *Source data 1* for the full sequence. Both alleles are homozygous viable with no apparent morphological phenotype. Both are fully inactivated by Flp-mediated excision of the FRT cassette. We also generated a reporter allele, $dpp^{FRT-REP}$, by inserting the DNA fragment shown in *Figure 3—figure supplement 1* in the *attP* site of the deletion allele used to generate $dpp^{FRT-CA}$ (see *Figure 1B*). In this construct, CD8-GFP coding sequences are located downstream of an HA-Dpp excisable cassette. See *Source data 1* for the full sequence. The $dpp^{FO}$ allele (*Akiyama and Gibson, 2015*), referred to here as $dpp^{FRT-TA}$ was obtained from Matt Gibson (Stower's Institute). Tubα1-FRT-f⁺-FRT-Dpp was described previously (*Zecca et al., 1995*). The other strains used for this study were obtained from the Bloomington stock centre. They include *rotund-Gal4 (rn-Gal4), nubbin-Gal4 (nub-Gal4), tubulin-Gal80^{ts}(II) (tub- Gal80^{ts})*, *UAS-Flp (X), hs-Flp (X)* and *hs-Flp (III)*.

## PCR analysis of genomic DNA

For Splinkerette PCR, DNA from single flies was isolated and digested with BglII. Afterwards, it was amplified following the Splinkerette PCR protocol for *Drosophila melanogaster* (*Potter and Luo, 2010*). Three $dpp^{BLK}$-Gal4 lines (which were kept in three labs for extended time) were analysed: $dpp^{BLK-TA}$-Gal4 (*Akiyama and Gibson, 2015*), $dpp^{BLK-CA}$-Gal4 (kept in London) and $dpp^{BLK-PSB}$-Gal4 (kept in Zürich). The following primers were used: SPLNK#1 + 5′SPLNK#1-GAWB for the first PCR round and SPLNK#2 + 5′SPLNK#2-GAWB for the second PCR round (see *Figure 4—source data 1* for primer sequences). The PCR products were isolated on a 2% agarose gel and sequenced with the primer 5′SPLNK-GAWB-SEQ. The size of the fragment differed for the three strains, probably because of polymorphism that accumulated during maintenance of the stocks. However, sequencing of the fragment showed that in all three cases, the insertion sites were identical, in the 5′UTR of CG6896 (MYPT-75D).

## qRT-PCR

Third instar larvae were heat shocked for 30 min at 102 hr AEL and wing discs were dissected in PBS at 120 hr AEL, before being transferred to PBS-Tween 20. Samples were spun down, and the pellets were snap-frozen in liquid nitrogen, stored at −80°C or processed immediately. RNA from the dissected discs was extracted with the Macherey-Nagel NucleoSpin RNA isolation kit, and cDNA was obtained with the Roche Transcriptor high fidelity cDNA synthesis kit. Quantitative PCR was performed in triplicates using the MESA Green qPCR Mastermix Plus for SYBR assay. All measurements were normalized to *actin-5C*, *alpha-tubulin* and *TATA box binding protein* mRNA levels. See *Figure 4—source data 1* for primer squences.

## Imaging

Imaginal discs were fixed in 4% paraformaldehyde for approximately 30 min before immunofluorescence staining. The following antibodies were used: α-Brinker (Aurelio Telemann, EMBL; 1/500), α-Brinker (Hillary Ashe, University of Manchester; 1/500); α-HA (Cell Signalling; 1/3000 or 1/500), α-Hth (Richard Mann, Columbia University; 1/500), α- Phospho-Histone H3 (Abcam; [HTA28] phospho S28; 1/500), α− Phospho-Smad1/5 (Cell Signalling; 41D10 #9516; 1/100)

α-DSRF (Active Motif; Cat 39093 Lot 03504001; 1/500), α-Omb (Gert Pflugfelder, University of Mainz; 1/500), and Alexa-conjugated secondary antibodies (Thermo Scientific Waltham, MA; 1/500). Images were acquired either with a Zeiss LSM710 or a Leica SP5 confocal microscope.

## Data analysis

Every experiment was repeated at least once. All data were analysed using Fiji (ImageJ) and Graph-Pad Prism. Error bars denote standard deviation (SD) unless stated otherwise, and the statistical tests used to evaluate significance are described in the figure legends. Statistical significance is denoted as follows: ns: $p > 0.05$, *$p \leq 0.05$, **$p \leq 0.01$, ***$p \leq 0.001$, ****$p \leq 0.0001$.

## Acknowledgements

This work was supported by core funding from the Francis Crick Institute and an advanced grant from the ERC (294523) to JP Vincent and grants from the Swiss National Science Foundation and the Canton of Zurich to K Basler. Ruta Ziukaite is the recipient of a PhD studentship from the Wellcome Trust. We thank Matthew C Gibson and the Bloomington Drosophila Stock Center for *Drosophila* strains. We also thank the colleagues (listed in Methods) who generously donated antibodies.

## Additional information

### Funding

| Funder | Grant reference number | Author |
| --- | --- | --- |
| Medical Research Council | FC001204 | Jean-Paul Vincent |
| European Research Council | WNTEXPORT 294523 | Jean-Paul Vincent |
| Schweizerischer Nationalfonds zur Förderung der Wissenschaftlichen Forschung | | Konrad Basler |
| Wellcome | PhD Studentship 105382/Z/14/Z | Ruta Ziukaite |
| Wellcome | FC001204 | Jean-Paul Vincent |
| Cancer Research UK | FC001204 | Jean-Paul Vincent |

The funders had no role in study design, data collection and interpretation, or the decision to submit the work for publication.

### Author contributions

PSB, RZ, Conceptualization, Formal analysis, Investigation, Methodology, Writing—original draft, Writing—review and editing; CA, Conceptualization, Formal analysis, Supervision, Investigation, Methodology, Writing—original draft, Writing—review and editing; KB, Conceptualization, Resources, Formal analysis, Supervision, Funding acquisition, Investigation, Writing—original draft; J-PV, Conceptualization, Resources, Supervision, Funding acquisition, Methodology, Writing—original draft, Project administration, Writing—review and editing

### Author ORCIDs

Pablo Sanchez Bosch, http://orcid.org/0000-0002-0574-4530
Ruta Ziukaite, http://orcid.org/0000-0001-7960-1834
Jean-Paul Vincent, http://orcid.org/0000-0003-2305-5744

## Additional files

### Supplementary files

• Source data 1. Allele sequences.

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
