## [Decision Letter]

Thank you for submitting your article "Dpp controls growth and patterning in *Drosophila* wing precursors through distinct modes of action" for consideration by *eLife*. Your article has been reviewed by three peer reviewers, and the evaluation has been overseen by a Reviewing Editor and K VijayRaghavan as the Senior Editor. The reviewers have opted to remain anonymous.

The reviewers have discussed the reviews with one another and the Reviewing Editor has drafted this decision to help you prepare a revised submission.

Summary:

A contentious item that continues to raise interest concerns the relationship between the gradient of the BMP4-like signaling protein Dpp produced in *Drosophila* wing discs and cell proliferation in the disc. Dpp is both necessary and sufficient for disc growth, and the problem basically boils down to why regions with different levels of Dpp and BMP signaling do not cause different amounts of growth. Evidence and arguments on this point remain of high current interest.

The present manuscript is a partial rebuttal to the 2015 Nature paper from Akiyama and Gibson, which used various types of *dpp* loss-of-function clones to argue that the BMP Dpp produced by the stripe of cells anterior to the A/P boundary in wing discs was not necessary for the growth of the disc. This argued that models based on reading a gradient of Dpp were likely wrong, and that levels could be greatly reduced without greatly affecting growth, and thus models based on a temporal gradient of increasing BMP signaling were probably wrong as well. Nonetheless, that study showed that the Dpp produced by the entire anterior compartment was necessary for growth, at least up until 36 hours before wandering third instar, presumably this was supplied by low-level Dpp produced outside the normal stripe of high level Dpp expression.

The present manuscript argue*s* that stripe Dpp is necessary for growth, using the same conditional *dpp-gal4* used by Akiyama to limit Dpp loss to the stripe, but different conditional *dpp* alleles that the authors suspect are more sensitive to recombinase. Thus, they argue that Akiyama got the wrong result because they did not remove *dpp* from the entire stripe at early enough stages. They also add some nice data that growth can be rescued by moderate levels of uniformly-expressed Dpp, which argues that a gradient it not needed for growth, even in the pouch where an endogenous gradient is found.

The authors have intelligently designed a conditional knock-out allele of Dpp wherein the FRT sites flank the final exon of Dpp and HA tag is inserted in the mature Dpp sequence to specifically label only mature Dpp. Upon expressing Flp using pouch specific GAL4s like rotund and nubbin, they observe that loss of Dpp in the pouch affects growth and that Dpp is continuously required throughout development. Unlike the results observed by Akiyama and Gibson, 2015, upon expressing FLP using the same *dpp-GAL4* driver, they indeed observe loss of Dpp and defects in growth. They show that the transgene generated by the Gibson group (dppTA) is not as efficiently excisable as an alternative transgene that they generate themselves (dppCA). They see a growth defect when this transgene is excised from the stripe at the same time as *brinker* is over expressed post-excision. The likely explanation is a positional effect of the FRT inserts and thus resulting in varying degrees of activity of FLP. This hypothesis has also been tested by expressing FLP using hs-FLP individually for the two conditional knockdowns and measuring the effect on phenotype (i.e. reduction in pouch size), consistent with Dpp emanating from the compartmental boundary is essential for growth. They provide important evidence using the Gal80/Gal4 system in the tub-driven *dpp* transgene in the back ground of the excised dppCA, that uniform tubulin promoter driven *dpp* is sufficient to promote growth, and repress *brinker*, but not to support patterning. Upon eliminating the endogenous stripe pattern of Dpp and introducing uniform low level expression of Dpp using *Tubα1/FRT/f+/FRT/dpp* transgene (Zecca et al., 1995) the authors provide evidence against a requirement of Dpp gradient for growth.

Essential revisions:

Reviewer 1:

1) One difficulty is that Akiyama show that their *dpp-gal4* technique removes most or all stripe Dpp from the dorsal wing pouch and hinge, and also greatly reduces pMad there, as early as 72 hours AEL (their Figure 3). Nonetheless, the dorsal pouch reaches a pretty normal-looking size by late third (although they did not measure pouch size alone, so it is possible there was a slight defect). If this is correct, then loss of the gradient and stripe do not affect growth from 72 hours on.

Either the authors need to disprove this, or they have to incorporate it into their Discussion. Does the Akiyama allele version of the experiment lead to loss of Dpp and the pMad gradient in parts of the disc at 72 hours, and is growth in those regions affected or unaffected?

If Akiyama is correct, this should be mentioned. One possible explanation is that the authors might investigate is that the early pMad loss was not enough to increase *brinker* expression at early time points, as Akiyama only examined *brinker* at late third. Since the authors have Akiyama's allele, could they look? My thinking here is that the different results might not be due to whether stripe Dpp is lost, per se, but how much residual Dpp signaling is left from Dpp elsewhere in the disc, and whether that residual signaling is enough to suppress *brinker* expression during the growth phase.

2) If dppCA works because it is more efficient, an important question is whether that removes stripe *dpp* at an earlier stage that dppTA, or whether it removes *dpp* over a broader region of the wing pouch. The authors seem to be suggesting the former, but the second raises the possibility that dppCA works because it is removing low-level Dpp from the far anterior of the wing disc. Both studies agree that widespread removal of Dpp reduces growth. *dpp-gal4* appears to have residual expression that can extend well into the anterior of at least the dorsal wing pouch (see the Evans G-TRACE paper, for instance). Unfortunately, none of the alleles have a way of checking where excision occurred, and comparison with some other FLPout construct would just lead to guesswork about which was more sensitive to FLPase. This is, for me, a critical question, and I have to hear some cogent counterargument to accept the authors' interpretation.

3) "This was confirmed by comparing side-by-side the impact of Flp expressed from a hsp70-flp transgene on dppFRT-TA and dppFRT-PSB". Since the *dpp* stripe experiment was done with dppCA, the critical issue is whether CA is more efficiently excised than TA. So why are the authors comparing TA with PSB? Do they also have the *dpp-gal4* data with PSB?

4) The graph in Figure 4 shows only a slight reduction PH3 after growth rescue compared with wild type, and the claim is that this shows rescue. But for this to be meaningful, there has to be a comparison with un-rescued, and data showing that PH3 is lower in un-rescued. That would also make a cleaner statistical comparison. Not significant is not the same as saying the numbers are the same, and cannot rule out a type II error, without a calculation of statistical power.

Reviewer 2:

1) Source of Dpp: As shown in Evan et al., 2009 using GTRACE, the cell lineage of *dpp* GAL4 not only marks the cells along the A/P boundary but also marks a portion of the anterior compartment. Although the results clearly demonstrate a that FLP under *dpp* GAL4 results in loss of mature form of *dpp* in the wing pouch and thus affects growth, the results do not rule out that a role for Dpp emanating from the compartmental boundary may also be necessary for growth.

2) Levels of Dpp: Although, using *Tubα1/FRT/f+/FRT/dpp* transgene (Zecca et al., 1995), the authors look at downstream effects of Dpp, it would be informative to know the levels and pattern of expression of Dpp in rescue and control discs. It would be useful to address when the de-repression of *brinker* in the context of the removal of dppTA takes place, compared to that when dppCA is excised. This will completely explain the discrepancy between the results obtained by the authors, here and the results of Akiyama and Gibson.

3) Anterior Dpp/Posterior Dpp: The above system could be used in conjunction with hhGAL4 or ciGAL4 driven FLP to remove endogenous Dpp within a compartment while still expressing uniform low levels of Dpp in the same compartment and verifying if growth/patterning is perturbed (related to Extended Data 6 of Akhiyama & Gibson et al., 2015). This will lay to rest an interesting proposition and provide a cautionary tale of (over) interpretation of the power of *Drosophila* genetic jugglery.

*Reviewer 3:*

1) At the beginning of results, it would be helpful for the reader to explain better the difference between the CA and PSB alleles (which remains a bit mysterious). Is the same cassette simply integrated in two different locations?

2) In the subsection “Temporal requirement of Dpp for wing growth” the authors note the local repression of *brinker* around the residual spots of Dpp expression (Figure 2', B'), however it is difficult to see this as currently displayed. It would be helpful to show the Brk channel alone.

3) In the subsection “Wing growth requires the endogenous stripe of Dpp expression” the authors conclude "Our findings so far indicate that Dpp must be produced in the pouch (prospective wing) during the 48-96h AEL period in order for the wing to grow." The fact that heat shock at 96h also gives reduced wing disc area (Figure 2) means that *dpp* is required for growth also after 96h at least for some time (although admittedly for how long post 96h AEL is not clear). But perhaps this conclusion could be rephrased slightly more strongly. (Since the cell cycle is roughly 12 hours at this point in development, and the reduction in size is almost 50%, does this imply Dpp is required for growth for at least another 12 hours – i.e. up to 108h AEL?)

4) Figure 3 simply shows in a different way that the effect of the PSB allele on growth is stronger than the effect of the TA allele, but it does not show it is due to "allele excisability”, or that it is more readily inactivated. This could be done, for instance, by performing Q-RT-PCR for Dpp transcript on the two alleles with mild heat shocks. Otherwise, the conclusion "that, the *dpp^FRT-TA^* allele may not be as readily inactivated as our alleles" should be rephrased.

The experiment removing endogenous Dpp and replacing it with a constitutively expressed Dpp under control of the tubulin promoter (Figure 4) is beautiful. It not only shows the requirement for Dpp for growth, but also further debunks the another publication (Wartlick et al. 2011) asserting that ever-increasing levels of Dpp are required for growth.

---

## [Author Response]

Essential revisions:

Reviewer 1:

1) One difficulty is that Akiyama show that their dpp-gal4 technique removes most or all stripe Dpp from the dorsal wing pouch and hinge, and also greatly reduces pMad there, as early as 72 hours AEL (their Figure 3). Nonetheless, the dorsal pouch reaches a pretty normal-looking size by late third (although they did not measure pouch size alone, so it is possible there was a slight defect). If this is correct, then loss of the gradient and stripe do not affect growth from 72 hours on.

Either the authors need to disprove this, or they have to incorporate it into their Discussion. Does the Akiyama allele version of the experiment lead to loss of Dpp and the pMad gradient in parts of the disc at 72 hours, and is growth in those regions affected or unaffected?

The genetic manipulation of Akiyama (with *dpp-gal4 UAS-Flp* and *dpp^FRT-TA^*) leads to loss of pMad from 72 h AEL (as shown in their paper). However, these authors did not assess Brinker expression at these early stages (only at the end of growth). We found that, in their genotype, Brinker is repressed at 90 h and 96 h AEL, a clear indication of residual Dpp signaling activity (see Figure 2). We suggest that, in their setup, *dpp* excision is incomplete, leaving a few cells in the stripe that produce Dpp at a level that is sufficient to inhibit Brinker, but not to activate pMad detectably (see more extensive discussion below). Since growth is relatively normal under these conditions, we conclude that Brinker repression, but not detectable pMad immunoreactivity, is needed for growth.

If Akiyama is correct, this should be mentioned. One possible explanation is that the authors might investigate is that the early pMad loss was not enough to increase brinker expression at early time points, as Akiyama only examined brinker at late third. Since the authors have Akiyama's allele, could they look? My thinking here is that the different results might not be due to whether stripe Dpp is lost, per se, but how much residual Dpp signaling is left from Dpp elsewhere in the disc, and whether that residual signaling is enough to suppress brinker expression during the growth phase.

We propose that the Akiyama allele is partially inactivated by *dpp-gal4 UAS-Flp*, leaving enough Dpp-expressing cells *within* the stripe for repression of Brinker during the growth period (but not enough to activate detectable pMad). Residual Dpp signalling is suggested by our observation that in *dpp-gal4 UAS-Flp dpp^FRT-TA^* discs, Brinker is repressed in the pouch at 90 and 96h AEL (Figure 2). We confirmed the observation of Akiyama that, at 120h AEL, Brinker is derepressed in this genotype (they only showed this time point). However, by that time, Dpp signalling is irrelevant for growth and therefore this observation erroneously suggests that growth can take place in the presence of Brinker. We suggest that, with the Akiyama allele, there is residual expression although there is no need to invoke another source of Dpp outside the stripe since inactivation of our alleles with *dpp-gal4 UAS-flp* does prevent growth. It is worth noting that low-level, uniform Dpp, which represses Brinker but fails to trigger detectable pMad, is sufficient for growth (Figure 4). Therefore, absence of detectable pMad immunoreactivity does not necessarily indicate complete loss of signaling activity.

2) If dppCA works because it is more efficient, an important question is whether that removes stripe dpp at an earlier stage that dppTA, or whether it removes dpp over a broader region of the wing pouch. The authors seem to be suggesting the former, but the second raises the possibility that dppCA works because it is removing low-level Dpp from the far anterior of the wing disc. Both studies agree that widespread removal of Dpp reduces growth. dpp-gal4 appears to have residual expression that can extend well into the anterior of at least the dorsal wing pouch (see the Evans G-TRACE paper, for instance). Unfortunately, none of the alleles have a way of checking where excision occurred, and comparison with some other FLPout construct would just lead to guesswork about which was more sensitive to FLPase. This is, for me, a critical question, and I have to hear some cogent counterargument to accept the authors' interpretation.

It is true that, as G-Trace shows, *dpp-Gal4* must have been active at some point during disc development far into the anterior compartment. It is reasonable to consider the possibility that such expression could inactivate the CA allele, but not the more ‘resistant’ TA allele in non-stripe anterior cells. Thus, remaining non-stripe Dpp expression from the TA allele (but not the CA or PSB alleles) would sustain growth in the presence of *dpp-Gal4 and UAS-Flp*. G-Trace integrates historical transcription activity, making it very sensitive but preventing an assessment of the level or timing of such activity in anterior cells. As an alternative, to determine the activity of the Dpp promoter, we have designed an allele (*dpp^FRT-REP^*) that carries FRTs in the same location as our original CA allele but in addition has sequences encoding CD8-GFP downstream of the FRT cassette such that, following excision, it expresses CD8-GFP under the control of the native *dpp* promoter. We have used this allele to map the transcriptional activity of *dpp* in the pouch by crossing it to *rotund-gal4 UAS-Flp* and staining the discs at 72, 96, and 120 h AEL. At all three stages, CD8-GFP was confined to the stripe (Figure 3—figure supplement 1). Since CD8-GFP is a sensitive reporter gene, we conclude that during the 72-120h period, the activity of the *dpp* promoter in the pouch is confined to the stripe. We suggest that the G-trace signal in anterior non-stripe cells could be due to earlier or very low promoter activity.

3) "This was confirmed by comparing side-by-side the impact of Flp expressed from a hsp70-flp transgene on dppFRT-TA and dppFRT-PSB". Since the dpp stripe experiment was done with dppCA, the critical issue is whether CA is more efficiently excised than TA. So why are the authors comparing TA with PSB? Do they also have the dpp-gal4 data with PSB?

In response to the reviewer’s comment, we have systematically compared the excisability of all three alleles by qRT-PCR. For each allele, excision was induced with the same *hs-Flp* transgene under identical heat shock conditions and the level of brinker expression in dissected discs was measured by qRT-PCR.

The results are consistent with the TA allele being less readily excised than the CA and PSB (Figure 3). As an additional test, we compared disc size at 120 h AEL for all three alleles following heat shocks at 72 or 96 h AEL. Following the 72 h heat shock, the PSB and CA discs were significantly smaller than the TA discs, showing that *dpp* was more thoroughly inactivated with our alleles than with TA (Figure 3).

4) The graph in Figure 4 shows only a slight reduction PH3 after growth rescue compared with wild type, and the claim is that this shows rescue. But for this to be meaningful, there has to be a comparison with un-rescued, and data showing that PH3 is lower in un-rescued. That would also make a cleaner statistical comparison. Not significant is not the same as saying the numbers are the same, and cannot rule out a type II error, without a calculation of statistical power.

As suggested, we have added data from un-rescued discs (Figure 4). Quantification shows that uniform expression of Dpp significantly restores proliferation (Figure 4).

Reviewer 2:

1) Source of Dpp: As shown in Evan et al., 2009 using GTRACE, the cell lineage of dpp GAL4 not only marks the cells along the A/P boundary but also marks a portion of the anterior compartment. Although the results clearly demonstrate a that FLP under dpp GAL4 results in loss of mature form of dpp in the wing pouch and thus affects growth, the results do not rule out that a role for Dpp emanating from the compartmental boundary may also be necessary for growth.

We assume that there is a typo in the reviewer’s comment and that he/she meant “a role for Dpp emanating *far* from the compartmental boundary may also be necessary for growth”. In our response to comment 2 of reviewer 1 (who also pointed out the G-Trace study), we describe results with a sensitive (but not permanent) reporter of *dpp* transcription that we created. They show that during the 72-120 h period, the sole domain of *dpp* expression in the pouch is the stripe along the A/P boundary (Figure 3—figure supplement 1).

2) Levels of Dpp: Although, using Tubα1/FRT/f+/FRT/dpp transgene (Zecca et al., 1995), the authors look at downstream effects of Dpp, it would be informative to know the levels and pattern of expression of Dpp in rescue and control discs. It would be useful to address when the de-repression of brinker in the context of the removal of dppTA takes place, compared to that when dppCA is excised. This will completely explain the discrepancy between the results obtained by the authors, here and the results of Akiyama and Gibson.

This is a two-part question, one about the level of Dpp expression from the *Tubα1/FRT/f+/FRT/dpp* transgene and the other about Brinker derepression in the context of the removal of *dpp^FRT-TA^*.

1) Unfortunately, Dpp produced from the *Tubα1/FRT/f+/FRT/dpp* transgene is not tagged and there is no suitable antibody to detect mature Dpp (the Akiyama antibody only recognizes the pro-domain). We have therefore assessed signaling activity in the rescue (and control) context with anti-P-Mad, anti-Omb, and anti-Brinker (Figure 4). The results show that *dpp* expressed from *Tubα1/FRT/f+/FRT/dpp* is too low to trigger detectable pMad, but sufficient to activate weak Omb expression and to repress Brinker. This is why we conclude that weak, low-level Dpp signaling is sufficient to promote growth.

2) As suggested by the reviewer we have compared the extent of Brinker derepression following inactivation of the various alleles. For all three alleles, qRT-PCR analysis shows that Brinker is derepressed following Flp expression from a *hs-Flp* transgene. Importantly however, this is less pronounced for TA than for CA or PSB (Figure 3). This was confirmed by immunofluorescence, comparing the three alleles (Figure 3). We conclude therefore that the TA allele is less readily inactivated. We also assessed the pattern of Brinker expression at 90, 96, and 120 h AEL in *dpp^FRT-TA^ dpp-Gal4 UAS-Flp* imaginal discs. The same assay was performed by Akiyama and Gibson but only at 120 h AEL. We confirmed their finding that Brinker is derepressed at 120 h, but found Brinker to be repressed at the earlier stages, when growth is taking place (Figure 2). By contrast, with the CA allele, expression of FLP from *dpp-Gal4 UAS-Flp* leads to early Brinker derepression and growth impairment (Figure 2) These observations are consistent with the previously accepted notion that Brinker must be repressed for the pouch to grow. As the reviewer recognizes, they also explain the discrepancy between our results and those of Akiyama and Gibson.

3) Anterior Dpp/Posterior Dpp: The above system could be used in conjunction with hhGAL4 or ciGAL4 driven FLP to remove endogenous Dpp within a compartment while still expressing uniform low levels of Dpp in the same compartment and verifying if growth/patterning is perturbed (related to Extended Data 6 of Akhiyama & Gibson et al., 2015). This will lay to rest an interesting proposition and provide a cautionary tale of (over) interpretation of the power of Drosophila genetic jugglery.

These are interesting suggestions but we wonder whether these experiments are necessary. All the evidence so far suggests that posterior deletion of Dpp (with *hh-Gal4 UAS-Flp*) will have no effect. With *ci-Gal4 and UAS-Flp*, it is very likely growth will be impaired since this is the case with the TA allele (which is less excisable than our allele). It would be fun to do a rescue experiment with *ci-Gal4* to test if flat expression in the A compartment is sufficient to promote growth. However, this experiment, which requires many generations, would significantly delay our paper.

Reviewer 3:

1) At the beginning of results, it would be helpful for the reader to explain better the difference between the CA and PSB alleles (which remains a bit mysterious). Is the same cassette simply integrated in two different locations?

This has been clarified. The FRT cassette is indeed at different locations for the three alleles. We note that one of the FRTs in the Akiyama allele is located right next to a LoxP site and wonder whether this might explain the lower excision efficiency.

2) In the subsection “Temporal requirement of Dpp for wing growth” the authors note the local repression of brinker around the residual spots of Dpp expression (Figure 2', B'), however it is difficult to see this as currently displayed. It would be helpful to show the Brk channel alone.

We are confident that local repression occurs but agree that it is not readily seen in the micrographs. Since this is not an important observation, we have deleted these panels and statement from the revised manuscript.

3) In the subsection “Wing growth requires the endogenous stripe of Dpp expression” the authors conclude "Our findings so far indicate that Dpp must be produced in the pouch (prospective wing) during the 48-96h AEL period in order for the wing to grow." The fact that heat shock at 96h also gives reduced wing disc area (Figure 2) means that dpp is required for growth also after 96h at least for some time (although admittedly for how long post 96h AEL is not clear). But perhaps this conclusion could be rephrased slightly more strongly. (Since the cell cycle is roughly 12 hours at this point in development, and the reduction in size is almost 50%, does this imply Dpp is required for growth for at least another 12 hours – i.e. up to 108h AEL?)

The reviewer rightfully notes that our data do not rigorously address whether Dpp is required for growth beyond 96 h. We cannot be sure for two reasons. One is that the growth rate decreases with age (as shown for example by Johnston and Sanders, NCB 2003), making any effect on growth harder to detect at later times. Second, we cannot exclude the possibility that perdurance of Dpp or downstream events might contribute to growth after excision. In light of these points, we have modified this section to indicate that Dpp signaling could very well contribute to growth until the end of larval stages.

4) Figure 3 simply shows in a different way that the effect of the PSB allele on growth is stronger than the effect of the TA allele, but it does not show it is due to "allele excisability”, or that it is more readily inactivated. This could be done, for instance, by performing Q-RT-PCR for Dpp transcript on the two alleles with mild heat shocks. Otherwise, the conclusion "that, the dpp^FRT-TA^ allele may not be as readily inactivated as our alleles" should be rephrased.

We agree that better evidence was needed. As suggested, for each allele, excision was induced with the same *hs-Flp* transgene under identical heat shock conditions and the level of residual *dpp* expression in dissected discs was measured by qRT-PCR. The results show higher residual expression for the TA allele than for the PSB and CA allele (Figure 3). We also assessed *brinker* expression by qRT-PCR under these conditions and the results are consistent with the TA allele being less readily excised than the CA and PSB (Figure 3). As an additional test, we compared the disc size at 120 h AEL for all three alleles following a heat shock at 72 or 96 h. The PSB and CA discs were consistently smaller than the TA discs, showing that Dpp was more thoroughly inactivated with PSB and CA than with TA (Figure 3).